# A Core Outcome Set to evaluate the impact of prognostication in people living with advanced cancer: An international consensus study

Caitlin Medlock[1¤]*, Bella Vivat[1☯], Nicola White[1☯], Jannicke Rabben[2‡], Patrick Stone[1☯]

**1** Marie Curie Palliative Care Research Department, Division of Psychiatry, University College London, London, United Kingdom, **2** Department of Health and Nursing Science, Faculty of Health and Sport Sciences, University of Agder, Kristiansand, Norway

☯ These authors contributed equally to this work.
‡ JR also contributed equally to this work.
¤ Current address: Cicely Saunders Institute of Palliative Care, Policy, and Rehabilitation, Florence Nightingale Faculty of Nursing, Midwifery & Palliative Care, King's College London, London, United Kingdom
* caitlin.medlock@kcl.ac.uk

## Abstract

### Background

Implementing a prognostic model in clinical practice requires assessing not just its accuracy but also how it may impact patient and healthcare outcomes. Evaluating these impacts requires clinical trials that are carefully designed to select, collect, and report key outcomes. However, there is currently no consensus on which outcomes are key to evaluating the impact of prognostication. Core Outcome Sets can standardise outcome selection, collection, and reporting, and thereby improve the comparability of studies. Our study aimed to develop a preliminary Core Outcome Set for evaluating the impact of prognostication on people living with advanced cancer.

### Methods

Following the Core Outcome Measures in Effectiveness Trials initiative's recommended methodology, we first conducted a systematic review of prognostic studies and in-depth interviews with patients, informal caregivers, and clinicians. Drawing on these findings, we produced a comprehensive list of potential outcomes, which we refined and prioritised through a two-round modified Delphi survey with an international expert panel. An online consensus meeting finalised the preliminary Core Outcome Set.

### Results

The systematic review identified 70 outcomes, and the interviews contributed another 16. Consolidation produced 67 outcomes for review in the Delphi survey. A total of 49

**Data availability statement:** All relevant data are within the paper and its Supporting information files.

**Funding:** A Marie Curie and the Economic and Social Research Council (ESRC) co-funded PhD Studentship grant was awarded to CM (grant number ES/P000592/1). We can confirm that the funders had no role in study design, data collection and analysis, decision to publish, or preparation of the manuscript. Marie Curie URL: https://www.mariecurie.org.uk/research-and-policy/research/calls-for-research ESRC URL: https://www.ukri.org/councils/esrc/guidance-for-applicants/types-of-funding-we-offer/.

**Competing interests:** The authors have declared that no competing interests exist.

experts from 14 countries; participated in the Delphi survey, with 31 (63%) completing both rounds. After two rounds, 34 outcomes were rated as critically important. These were discussed further in the consensus meeting, attended by 12 participants, who reached agreement on 9 core outcomes spanning five domains: physiological/clinical outcomes, global quality of life, care delivery, perceived health status, and personal circumstances.

## Conclusions

This preliminary Core Outcome Set offers a standardised approach for selecting, collecting, and reporting outcomes in prognostic impact studies in advanced cancer, which could enhance the comparability of future prognostic impact studies, and thereby potentially improve clinical practice and patient care outcomes.

## Introduction

Predicting length of survival (prognostication) has a critical role in managing advanced cancer; informing treatment decisions and care planning [1], influencing hospice transfers, and enrolment into integrated care pathways [2,3] and helping patients and their informal caregivers plan for the end-of-life [4]. Clinicians can predict survival in people living with advanced cancer in various ways, from subjective clinical judgment to prognostic models, which have been developed to provide more objective and reproducible estimates [5]. Examples of such models include the Palliative Prognostic Score (PaP) [6], the Palliative Prognostic Index (PPI) [7], and the Prognosis in Palliative Care Study (PiPS) models [8]. No single prognostic method is currently considered superior in terms of accuracy; studies have shown that they have similar levels of accuracy when compared with each other, and also with clinicians' own predictions [8–11]. This makes it difficult for clinicians to choose between prognostic methods and offers little incentive to use prognostic models in practice.

However, the effectiveness of a prognostic model in clinical settings is not determined solely by its accuracy. Different prognostic models can offer additional benefits, or pose potential harms that could impact patient care and decision-making, and these should all be considered, particularly when existing prognostic methods have broadly equivalent accuracy [12]. Clinical trials known as prognostic impact studies, which compare the real-world implications of various prognostic approaches, can evaluate the impact of prognostic models on patient and healthcare outcomes [12], such as quality of communication, or alignment of care with patient preferences, and assess whether using prognostic models in clinical practice leads to measurable and meaningful differences in clinically important outcomes [12]. However, no such studies have yet been conducted in the context of advanced cancer. Compounding this issue is the absence of standardised outcomes in relevant studies, which complicates the selection, collection, and reporting of outcomes necessary for comparing the impact of different prognostic models. Therefore, there is a pressing need for a

standardised set of outcomes that can be uniformly selected, collected and reported in prognostic impact studies involving people living with advanced cancer.

The Core Outcome Measures in Effectiveness Trials (COMET) initiative offers a solution to this problem by advocating for the use of a Core Outcome Set (COS) [13]. A COS defines a minimum set of outcomes that should be consistently measured and reported across all studies within a particular field, thereby standardising outcome selection, collection, and reporting, and enhancing comparability between studies [14]. Developing a COS to evaluate the impact of prognostication in advanced cancer would facilitate future prognostic impact studies and enable the assessment of whether prognostic models offer advantages beyond accuracy that support their use in clinical practice. This, in turn, could contribute to improving patient care by informing more effective, evidence-based prognostication strategies, ultimately leading to better-informed decision-making for both patients and clinicians.

This paper presents the findings from a three-phase study designed to develop a COS to evaluate the impact of prognostication in people living with advanced cancer. This project focused on outcomes for use in comparative impact studies, such as randomised controlled trials and other comparative study designs. The study had three objectives:

1. To develop a list of outcomes relevant to the impact of prognostication on people living with advanced cancer.

2. To prioritise outcomes important to relevant stakeholders, namely patients, informal caregivers, and clinicians, for measuring the impact of prognostication in advanced cancer.

3. To achieve consensus on a minimum set of core outcomes to evaluate the impact of prognostication in people living with advanced cancer.

This paper is reported in accordance with the Core Outcome Set-STAndards for Reporting (COS-STAR) guidelines [15].

## Materials and methods

### Protocol/registry entry

The research protocol for this study was registered with the COMET (Core Outcome Measures in Effectiveness Trials) initiative (COMET ID: 2136) (https://www.comet-initiative.org/Studies/Details/2136) and published *a priori* in an open-access journal [16]. The development of the COS followed a three-phase process (Fig 1) using methodology endorsed by COMET [14] and adhering to the Core Outcome Set-STAndards for Development (COS-STAD) minimum criteria [17]. The aims of each Phase are described briefly in Fig 1. The methods for each step can be found in more detail in the study protocol [16].

### Information sources

We identified a long list of candidate outcomes by conducting a systematic literature review of both quantitative and qualitative studies (Phase I) [18], and in-depth interviews with people living with advanced cancer, informal caregivers, and clinicians (Phase II) [19]. These results laid the groundwork for Phase III.

**Systematic review (Phase I).** Phase I aimed to identify existing outcomes in published studies to inform outcome and outcome domains to be included in the COS. We conducted a systematic review in accordance with PRISMA guidelines [20]. The review protocol was registered with PROSPERO (CRD42022320117, 29/03/2022) and has been published, with the full findings of the review reported elsewhere [18]. We systematically searched five electronic databases (MEDLINE, Embase, PsycINFO, CINAHL, and the Cochrane Library) from database inception to March 2022, using a comprehensive strategy developed with input from an information specialist. Studies were eligible for inclusion if they involved adults (aged ≥18 years) with advanced cancer or their informal caregivers, and if they reported on outcomes related to the impact of prognostication or explored relevant experiences and perceptions. For the purposes of this review, "impact" referred to the effects of prognostication on patients, informal caregivers, or clinicians, including clinical, symptomatic,

**Phase I: Systematic review**

- Identify outcomes used to assess the impact of prognostication in advanced cancer in the extant quantitative literature.

- Synthesise qualitative research exploring patients' and informal caregivers' experiences of prognostication in advanced cancer.

- Create a preliminary list of outcomes and outcome domains to potentially be included in the COS.

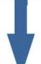

**Phase II: Semi-structured interviews**

- Identify outcomes of prognostication that are important to stakeholders.

- Compare outcomes identified from the interviews with those found in the literature to supplement the list of outcomes identified in Phase I for consideration in the COS.

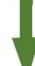

**Phase III: Consensus process**

- Create a comprehensive long list of outcomes from those identified in Phases I and II to inform an international, online, two-round, modified Delphi survey.

- Rate these outcomes according to the perspectives of patients, informal caregivers, clinicians, and academics/researchers using an international, online, two-round, modified Delphi survey.

- Obtain consensus on a minimum set of the most 'core' outcomes for evaluating the impact of prognostication in people living with advanced cancer.

**Fig 1. Phases of developing a COS to evaluate the impact of EOL prognostication in people living with advanced cancer.**

psychological, and service-level outcomes. We included both quantitative and qualitative study designs. Studies were excluded if they focused solely on prognostic model development or validation without assessing the impact of prognostication on individuals, or if they involved non-cancer populations. Quantitative outcome data were extracted and reported by prevalence, capturing clinical effects, symptomatic and psychological impacts, and service-level indicators (e.g., waiting list durations, bed occupancy rates). Qualitative data were thematically synthesised, capturing terminology and themes relevant to the prognostication experience. All extracted outcomes were systematically categorised into domains using a modified COMET taxonomy [13], with an additional domain for spiritual/religious/existential functioning/wellbeing incorporated.

**Semi-structured interviews (Phase II).** In Phase II, we conducted semi-structured interviews exploring the experiences of prognostication from the perspectives of key stakeholders. Interviews were conducted with 8 people living with advanced cancer, 10 informal caregivers, and 10 clinicians, recruited from two NHS sites and one non-NHS hospice in London, United Kingdom (UK). These interviews took place between January and September 2023, either in person, via telephone, or via video conferencing. All interviews were transcribed and analysed using framework analysis. During analysis, transcripts were coded for experiences and perceptions relating to prognostication, with particular attention to consequences or effects that participants described as important. These were interpreted as potential outcomes. The outcomes identified from these interviews were then mapped to the modified COMET taxonomy [13], which served as the analytical framework, and compared with outcomes generated by the systematic review to identify any novel outcomes not previously captured in the literature (see S1 File for a full list of outcomes identified in each Phase). The full findings from this phase have been published and are reported elsewhere [19].

## Consensus process (Phase III)

**Creating a long list of outcomes.** The lead author (CM) consolidated the outcomes extracted from the systematic review and interviews, removing overlapping terms, merging similar concepts, and generating additional outcomes where necessary. The co-authors and a Study Advisory Group (SAG) (see further details below), then reviewed this revised long list. Draft working definitions for the outcomes were discussed and refined within the SAG, and subsequently reviewed by the central research team to ensure consistency, clarity, and accessibility of the terminology used.

**Delphi survey.** The final long list of outcomes informed the development of an international, online, two-round, modified Delphi survey (Phase III). The survey was designed and hosted on the online software platform *REDCap* [21]. The aim was to achieve consensus on a minimum set of the most important outcomes for evaluating in the impact of prognostication in people living with advanced cancer. Participants rated each outcome's importance using a 9-point Likert scale and were encouraged to suggest additional outcomes they felt were missing. Only those who completed Round 1 were invited to participate in Round 2. In the second round, participants received the average ratings for each outcome across stakeholder groups, alongside their individual ratings. They were then prompted to reflect on both their initial ratings and the group averages before re-evaluating each outcome. The final scores for outcomes to be considered at the consensus meeting were calculated based solely on the data from participants who completed both rounds.

To assess participant retention and engagement throughout the study, we calculated the overall attrition and retention rates for each stakeholder as percentages for each round and across both rounds [22]. The potential for attrition bias was explored using the methods of Harman et al. [23]. We calculated the average score across all outcomes from round 1 for each participant and then compared these averages between participants who completed both rounds and those who only completed round 1. To visualise the distribution of these average ratings, we used bar charts to compare the distribution of the two groups' average ratings, allowing for a clear comparison of potential differences in perspectives.

**Consensus meeting.** Upon completing round 2 of the Delphi survey, participants were invited to a consensus meeting aimed at agreeing on the core outcomes to be included in the COS, conducted online to facilitate international participation. The lead author (CM) led the session, with a co-author (JR) serving as a facilitator to support the discussion, ensure that all participants had opportunities to contribute, and help prevent any one stakeholder group from dominating the conversation. During the meeting, outcomes rated as critically important in the Delphi survey were discussed for potential inclusion in the COS. A round robin approach was used to ensure each participant had the opportunity to speak during these discussions. Participants used the same Likert scale from the Delphi survey to rate each outcome's importance through *Mentimeter* for anonymous live polling. Outcomes that did not achieve consensus in the Delphi survey were also addressed; if participants identified these as important, they were included in the rating exercise. Outcomes voted as critically important were incorporated into the final COS.

**Participants.** We recruited individuals from four stakeholder groups: people living with advanced cancer, informal caregivers, clinicians, and academics/researchers. The eligibility criteria, recruitment strategy, and sample size are detailed further in the study protocol [16]. Recruitment took place from 22nd January until 30th April 2024.

**Outcome scoring.** We asked participants to rate the importance of each outcome using the Grading of Recommendations Assessment, Development, and Evaluation (GRADE) scale [24]. This scale consists of a nine-point numerical range with descriptive categories: scores of 1–3 indicate limited importance, 4–6 denote that the outcome is important but not critical, and 7–9 signify that the outcome is critically important. We also included an 'unable to score' option, so participants could indicate if they could not reach a decision on an outcome.

**Consensus definition.** According to COMET guidelines [14], for an outcome to be included in a consensus meeting, ≥70% of participants need to rate it as critically important (7–9 on the scale), and ≤15% rate it as of limited importance [1–3]. Conversely, consensus for exclusion requires ≥70% of participants to rate the outcome as of limited importance and ≤15% to rate it as critically important. Outcomes that do not meet either criterion are considered not to have reached consensus.

Only outcomes that met the exclusion criteria were dropped after the first round; all other outcomes, whether meeting the inclusion criteria or lacking consensus, were retained for the second round. To ensure the perspectives of individual stakeholder groups were adequately represented, any outcome rated as critically important by at least one stakeholder group, even if it met the criteria for exclusion, was retained for further consideration in the subsequent round.

After the two rounds of the Delphi survey, outcomes rated as critically important were shortlisted for discussion in the consensus meeting. Outcomes that had not reached consensus were compiled into a separate list, with those outcomes which participants considered important being available for discussion during the meeting.

During the consensus meeting, only those outcomes that met the criteria for inclusion during the rating exercise were included in the COS. Table 1 summarises the consensus criteria and actions used.

**Ethics and consent.** Ethical approval for the study was granted on 6th September 2022 by the London-Camberwell St. Giles Research Ethics Committee (reference 22/LO/0469). Written, informed consent was collected electronically from all participants at the beginning of each round of the Delphi survey and prior to the consensus meeting via *REDCap*. All Delphi survey responses were anonymised, with no identifiable participant data collected. While the consensus meeting was not anonymous, participant contact and logistical arrangements were managed via *REDCap*, ensuring that the research team did not have access to participants' personal contact information.

**Patient and public involvement.** In line with COMET guidelines [14], a SAG was established at the outset of this study, consisting of public research partners experienced in discussing prognosis in advanced cancer contexts. This group comprised of five individuals with lived experience of advanced cancer: two people living with advanced cancer, two bereaved informal caregivers, and one current informal caregiver. All members had experience of discussing prognosis in advanced cancer contexts, and were involved from the study's inception to its dissemination. Their involvement was crucial in co-producing the research, as they guided various aspects, including the design, recruitment, development, and dissemination of the COS. This collaboration ensured that the research remained patient-centred and that the outcomes selected for the final COS aligned with the priorities and lived experiences of those affected by advanced cancer.

## Results

### Participant characteristics

In all, 49 individuals (10 people living with advanced cancer, 11 informal caregivers, 18 clinicians, and 10 academics/researchers) from 14 countries completed round 1 of the Delphi survey. Thirty-one (4 people living with advanced cancer, 5 informal caregivers, 15 clinicians, and 7 academics/researchers) also completed round 2. Table 2 outlines the demographic characteristics of participants across each stage of the consensus process.

**Table 1. Consensus criteria and actions.**

| Consensus status | Criteria | Action |
|---|---|---|
| **Delphi survey** | | |
| Inclusion | ≥70% rate the outcome as critically important (7–9) **AND** ≤15% rate the outcome as of limited importance (1–3) | The outcome is taken forward to round 2. At the end of round 2, the outcome is discussed in the consensus meeting and has the potential to be included in the final COS. |
| Exclusion | ≥70% rate the outcome as of limited importance (1–3) **AND** ≤15% rate the outcome as critically important (7–9) **AND** No stakeholder group rates the outcome on average as critically important (7–9) | The outcome is removed from round 2 of the Delphi survey and is not included in the COS. |
| No consensus | The outcome does not meet the above criteria | The outcome remains in round 2. There will be an opportunity to discuss the outcome in the consensus meeting. |
| **Consensus meeting** | | |
| Inclusion | ≥70% rate the outcome as critically important (7–9) **AND** ≤15% rate the outcome as of limited importance (1–3) | The outcome is included in the COS. |
| Exclusion | The outcome does not meet the above criteria | The outcome is not included in the COS. |

## Delphi survey

From Phases I and II, 86 unique outcomes were identified. After consolidation and review by the SAG and research team, a refined list of 67 outcomes, organised into 13 outcome domains of the modified COMET taxonomy, was carried forward to the Delphi survey in Phase III (Table 3).

Of the 67 outcomes, 20 met the criteria for inclusion across all participants in the first round of the Delphi survey, while none met the criteria for exclusion, and no consensus was reached on the remaining 47 (see S2 File). Participants provided 15 comments regarding potential outcomes. Based on these suggestions, one new outcome (physical functioning) was added to round 2, and wording for two existing outcomes was revised. *Spiritual crisis* was revised to *spiritual/religious/existential crisis* to encompass the diverse ways individuals may experience crises related to beliefs, purpose, and existential concerns [25]. Additionally, *patient-doctor relationship* was revised to *patient-clinician relationship* to reflect the diverse roles within the care team, aligning with contemporary healthcare practices that emphasise the importance of recognising all healthcare providers in the therapeutic relationship [26].

In round 2, a total of 68 outcomes were rated. Of these, 34 met the criteria for inclusion and were taken forward to the consensus meeting, while the remaining 34 did not reach consensus (see S3 File). The attrition rate between rounds 1 and 2 was 37%, with 31 out of 48 participants remaining. Patients had the highest attrition rate (60%), followed by informal caregivers (55%), academics/researchers (30%), and clinicians (17%).

**Table 2. Participant characteristics.**

| Participant characteristics | Delphi survey round 1 (*n*=49) | Delphi survey round 2 (*n*=31) | Consensus meeting (*n*=12) |
|---|---|---|---|
| **Stakeholder group, n (%)ᵃ** | | | |
| Patients | 10 (20) | 4 (13) | 2 (17) |
| Informal caregivers | 11 (22) | 5 (16) | 2 (17) |
| Clinicians | 18 (37) | 15 (48) | 5 (42) |
| Academics/researchers | 10 (20) | 7 (23) | 3 (25) |
| **Age category (years), n (%)** | | | |
| 18–25 | 1 (2) | – | – |
| 26–35 | 5 (10) | 2 (6) | 2 (17) |
| 36–45 | 8 (16) | 6 (19) | 3 (25) |
| 46–55 | 14 (29) | 11 (35) | 4 (33) |
| 56–65 | 14 (29) | 6 (19) | 3 (25) |
| 66–75 | 6 [12] | 6 (19) | – |
| 76–85 | 1 (2) | – | – |
| **Gender, n (%)** | | | |
| Male | 18 (37) | 11 (35) | 6 (50) |
| Female | 28 (57) | 18 (58) | 6 (50) |
| Prefer to self-describe | 1 (2) | – | – |
| Prefer not to say | 2 (4) | 2 (6) | – |
| **Ethnicity, n (%)** | | | |
| Arab | 1 (2) | 1 (3) | – |
| Asian (Chinese) | 2 (4) | 1 (3) | – |
| Asian (Other) | 3 (6) | 3 (10) | 3 (25) |
| Black (African) | 3 (6) | 1 (3) | 1 (8) |
| Mixed/multiple | 3 (6) | 1 (3) | 1 (8) |
| White (British) | 15 (31) | 9 (29) | 4 (33) |
| White (Irish) | 1 (2) | 1 (3) | – |
| White (Other) | 18 (37) | 11 (35) | – |
| White (Not specified) | 1 (2) | 1 (3) | 3 (25) |
| Other (Not specified) | 1 (2) | 1 (3) | – |
| Prefer not to say | 1 (2) | 1 (3) | – |
| **Country of residence, n (%)** | | | |
| Australia | 1 (2) | 1 (3) | – |
| Belgium | 1 (2) | 1 (3) | – |
| Canada | 2 (4) | 2 (6) | – |
| England | 27 (55) | 18 | 7 (58) |
| Italy | 1 (2) | 1 (3) | – |
| Japan | 2 (4) | 2 (6) | 2 (17) |
| Kenya | 1 (2) | – | – |
| Portugal | 1 (2) | – | – |
| Scotland | 1 (2) | – | – |
| Singapore | 1 (2) | 1 (3) | 1 (8) |
| Sweden | 3 (6) | 1 (3) | – |
| Switzerland | 3 (6) | 1 (3) | – |
| USA | 4 (8) | 3 (10) | 2 (17) |

*(Continued)*

Table 2. (Continued)

| Participant characteristics | Delphi survey round 1 (*n*=49) | Delphi survey round 2 (*n*=31) | Consensus meeting (*n*=12) |
|---|---|---|---|
| Wales | 1 (2) | – | – |
| **Highest level of education, n (%)** | | | |
| No qualifications | 1 (2) | – | – |
| Secondary education | 3 (6) | 2 (6) | 1 (8) |
| Higher education below degree | 1 (4) | 1 (3) | – |
| Degree | 3 (6) | 4 913) | 1 (8) |
| Postgraduate | 33 (67) | 24 (77) | 10 (83) |
| Prefer not to say | 1 (2) | – | – |

USA: United States of America.

ᵃNot all percentages add up to 100% owing to rounding.

To assess potential attrition bias, we compared the average ratings for each outcome in round 1 between participants who completed both rounds (*n*=31) and those who completed only round 1 (*n*=18). Bar charts in S4 File visually represent these comparisons, with blue, shaded bars indicating participants from round 1 only and white, open bars indicating those who completed both rounds. In the absence of attrition bias, the distributions should overlap, showing similar patterns.

The bar charts confirm that the average ratings of participants who completed both rounds were closely aligned with those of participants who only completed round 1, suggesting minimal attrition bias. Further analyses by stakeholder group showed consistent patterns, indicating that participants who dropped out after round 1 generally shared similar views with those who completed both rounds.

## Consensus meeting

16 people were expected to attend the consensus meeting, but in the event 12 people from 4 countries did so (2 people living with advanced cancer, 2 informal caregivers, 5 clinicians, 3 academics/researchers). During the meeting, the 34 outcomes that met the inclusion criteria in the second round of the Delphi survey were presented and discussed. The other 34 outcomes that had failed to reach consensus were also reviewed. However, none of these were considered important enough to include in the rating exercise. Through a single round of live rating at the meeting, consensus was achieved on nine core outcomes (see S5 File), which were then established as a provisional COS (see Table 4). Fig 2 illustrates the outcome selection process throughout the study.

## Discussion

This paper reports the findings of a three-phase study aimed at developing the first provisional COS to evaluate the impact of prognostication in people living with advanced cancer. The study involved collaboration with an international group of key stakeholders consisting of patients, informal caregivers, clinicians, and academics/researchers. Consensus was reached on nine core outcomes, consistently rated as critically important by at least 70% of participants across all stages of the consensus process. These outcomes are physical functioning, psychological/mental status, quality of life, treatment/care preferences, end-of-life/advance care planning, place of care, quality of death, prognostic understanding, and practical/logistical preparation for end-of-life. These nine outcomes represent a recommended minimum set that should be selected, collected, and reported in future prognostic impact studies in advanced cancer to improve consistency and comparability across research.

**Table 3. Long list of outcomes used to inform the Delphi survey.**

| Outcome domains (*n*=13) | Outcomes (*n*=67) |
|---|---|
| Mortality/survival | Length of survival |
| Physiological/clinical outcomes | Pain |
| | Drowsiness |
| | Nausea |
| | General malaise |
| | Weakness |
| | Breathlessness |
| Psychiatric outcomes | Depression |
| | Anxiety |
| | Psychological/mental status |
| | Psychological distress |
| Spiritual/religious/ existential functioning/wellbeing | Spectrum of hope |
| | Being at peace with dying |
| | Spiritual and religious coping |
| | Spiritual crisis |
| | Loss of interest/pleasure |
| | Loss of resilience |
| | Loss of dignity |
| | Dissatisfaction with life |
| | Perceived sense of burden on others |
| | Sense of suffering |
| | Sense of control |
| | Desire for death |
| | Wish to live |
| | Worry about dying |
| Emotional functioning/wellbeing | Disbelief, shock, and denial |
| | Avoidance of prognosis |
| | Prognostic acceptance |
| | Emotional distress |
| | Use of coping strategies/mechanisms |
| | Fixation on prognosis |
| | Mental/emotional preparation for end-of-life |
| | Achieving/prioritising personal goals and values |
| | Anticipatory grief in patients |
| | Anticipatory grief in informal caregivers |
| | Having the opportunity to say goodbye to loved ones |
| | Decisional satisfaction |
| | Regret in informal caregivers |
| | Bereavement in informal caregivers |
| Cognitive outcomes | Cognitive function |
| Social functioning | Quality of communication between patient and family/friends |
| | Quality of patient-informal caregiver relationship |
| | Quality of relationships with others |
| | Social isolation |

*(Continued)*

**Table 3.** (Continued)

| Outcome domains (*n*=13) | Outcomes (*n*=67) |
|---|---|
| Global quality of life | Quality of life |
| Delivery of care | Treatment/care preferences |
| | Shared decision making |
| | End-of-life/advance care planning |
| | Information needs/preferences |
| | Patient-doctor relationship |
| | Family informed about imminent death |
| | Family present at time of death |
| | Place of care |
| | Place of death |
| | Quality of death |
| | Access to practical support |
| | Access to financial support |
| | Participation in clinical trials/research |
| Perceived health status | Prognostic awareness |
| | Prognostic understanding |
| | Being aware of prognostic uncertainty |
| Personal circumstances | Practical/logistical preparation for end-of-life |
| | Financial concerns |
| Hospital/hospice use | Hospice enrolment |
| | Admission to hospital |
| | Length of hospital admission |
| Societal/carer burden | Informal caregiver/family challenges |

**Table 4. Provisional Core Outcome Set to evaluate the impact of prognostication in people living with advance cancer.**

| Outcome domain | Outcome |
|---|---|
| Physiological/clinical outcomes | Physical functioning |
| | Psychological/mental status |
| Global quality of life | Quality of life |
| Delivery of care | Treatment/care preferences |
| | End-of-life/advance care planning |
| | Place of care |
| | Quality of death |
| Perceived health status | Prognostic understanding |
| Personal circumstance | Practical/logistical preparation for end-of-life |

The inclusion of outcomes like quality of life, physical functioning, and treatment preferences aligns with findings from other COS studies in advanced cancer populations [27–29]. This consistency suggests that these outcomes are universally pertinent across different interventions and care settings, reinforcing their significance in evaluating patient care. Conversely, this COS does not include some outcomes traditionally prioritised in prognostic and advanced cancer research, such as length of survival and specific symptom-related outcomes (e.g., pain, fatigue, and emotional

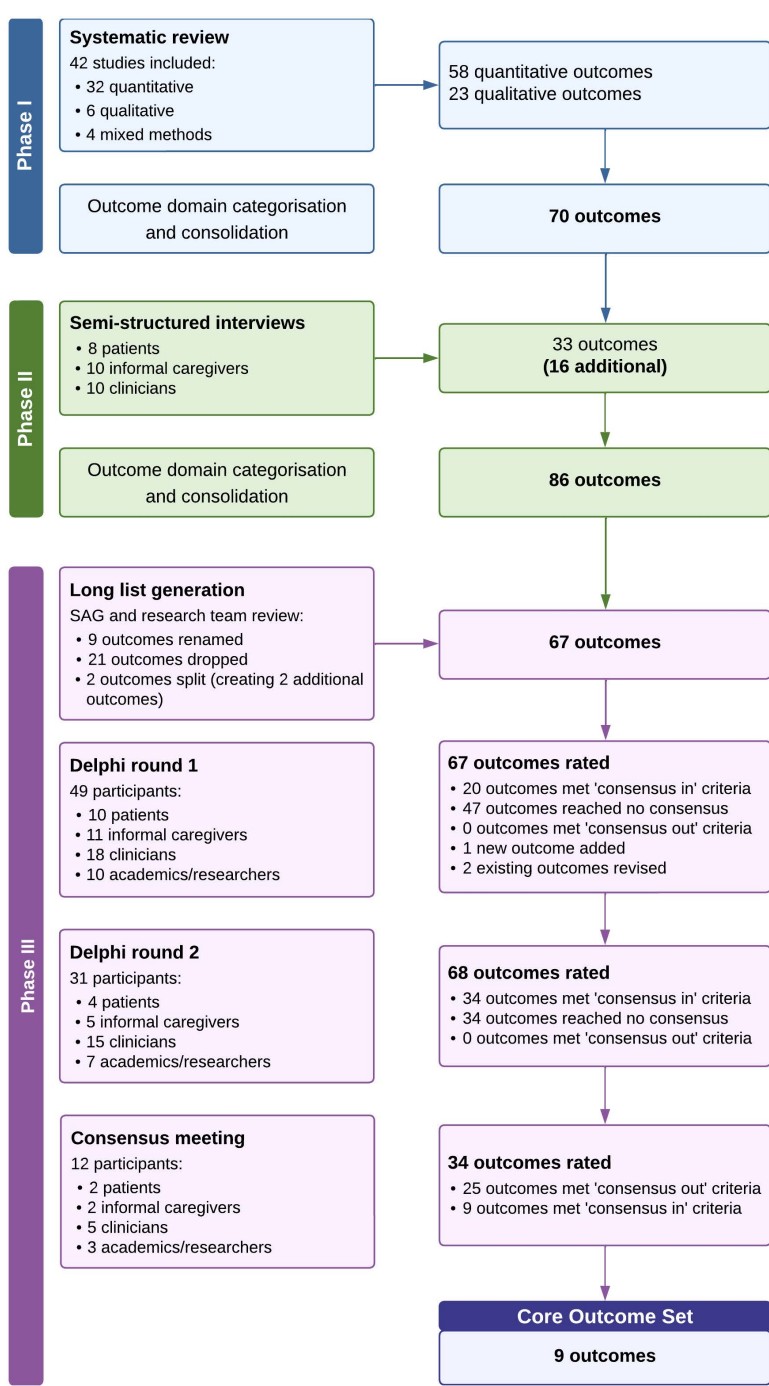

**Fig 2. Flowchart of outcome selection throughout the study.**

distress) [28–31]. The exclusion of survival reflects the COS's focus on outcomes directly influenced by prognostic communication, rather than those shaped by broader clinical or biological factors. Since survival is already a routine endpoint in prognostic studies, including it could introduce redundancy and distract from the evaluation of more subjective, care-related impacts. The absence of symptom-related and psychosocial outcomes may also stem from the perception that these are encompassed within the broader domain of quality of life. However, this raises concerns about the potential underrepresentation of these important dimensions, particularly given that they were rated as important in earlier phases of the study. Furthermore, the limited diversity of participants, and especially the underrepresentation of patients and informal caregivers during the final consensus meeting, may have influenced outcome prioritisation. A more diverse stakeholder group may have identified different or additional priorities. For these reasons, this COS should be regarded as provisional, and future research should seek to validate and refine it through broader stakeholder engagement.

It is equally important to emphasise that the COS is not intended to be restrictive. In line with COMET guidance [14], it provides a minimum set of outcomes to ensure standardisation, but researchers are strongly encouraged to include additional outcomes that are relevant to their specific study aims. The exclusion of outcomes related to survival, symptoms, or emotional, social, and spiritual aspects does not imply that they are unimportant, only that they may not be essential in every prognostic impact study.

In addition to identifying *what* to measure, the long-term success of this COS depends on establishing consensus on *how* and *when* these outcomes should be measured [14]. As highlighted in our earlier systematic review, considerable variability exists in both outcome definitions and measurement instruments [18]. Further work is currently underway to determine appropriate tools and optimal timepoints for each outcome to support consistent implementation and facilitate meaningful synthesis across studies.

Despite these limitations, this COS represents a significant advancement toward standardising outcome selection in prognostic impact research. Its use has the potential to improve the quality and comparability of research findings, reduce waste, and enable meta-analyses and other forms of evidence synthesis. Beyond its value in research, the COS may also support the development of evidence-based clinical guidelines. Standardised outcome data could guide clinicians, including oncologists, palliative care physicians, and clinical nurse specialists, in delivering more consistent, clear, and person-centred prognostic communication. This consistency is essential to ensuring equitable access to reliable prognostic information across healthcare settings, ultimately contributing to better-informed patients, improved care coordination, and more effective clinical decision-making.

## Strengths and limitations

This study was conducted and reported in adherence to established and internationally accepted COMET methodology [14,15], and included public research partners throughout the research process. A strength of the Delphi survey was the lack of attrition bias, despite the low retention rate. Additionally, holding a joint consensus meeting offered the potential for each stakeholder group to hear and consider others' perspectives, facilitating open discussion and the prioritisation of outcomes most relevant to those who will ultimately benefit from the COS.

A major limitation of this study is the limited representation of patients and informal caregivers in the consensus meeting. The numbers of these participants were disproportionately small compared to the total numbers across all stages of the process. This imbalance is particularly important to acknowledge, as patients and informal caregivers bring unique and critical insights into the relevance and importance of certain outcomes, insights that may differ significantly from those of clinicians and researchers [32,33]. The limited representation of these stakeholder groups may have unintentionally reduced the weight given to outcomes most meaningful to them. Consequently, this could affect the extent to which the provisional COS reflects the priorities of all stakeholder groups, particularly those who are most directly impacted by prognostication in advanced cancer.

There were also limitations regarding the lack of diversity among participants across the entire consensus process. The majority of the participants were White and resided in the UK. This demographic skew also raises concerns about the content and the external validity and generalisability of this provisional COS, because the perspectives of non-white populations and those from outside the UK are underrepresented. In addition, the international representation in the study was limited to participants from high- and middle-income countries. This is especially concerning as the needs and experiences of patients in low-income settings may differ significantly from those in high- and middle-income countries. Such disparities indicate that this provisional COS may not adequately reflect or address the challenges faced by populations in lower-income countries, highlighting a critical gap that needs addressing in future research.

The study also demonstrated limited diversity in educational levels, with most participants in all four stakeholder groups possessing at least a degree, if not a postgraduate qualification. Educational attainment is known to influence participation in health surveys, where people with lower levels of education are often underrepresented [34]. This underrepresentation may stem from lower awareness or perceived relevance of certain topics, such as health [35]. The predominance of higher education among participants in this study may have also skewed the outcomes, over-representing views commonly held by people with more formal education and potentially neglecting the unique insights or needs of others. This lack of educational diversity risks narrowing consensus on outcomes that do not fully account for experiences across all communities.

## Conclusion

Our study achieved international consensus on nine core outcomes across five outcome domains that may be important to select, collect, and report in prognostic impact studies in advanced cancer. While the COS represents an important step toward improving consistency in outcome measurement and reporting, it should be viewed as a provisional framework. Its use in future prognostic studies has the potential to enhance the comparability of findings and promote more person-centred research by focusing on outcomes that matter to key stakeholders. However, further research is needed to identify appropriate outcome measurement instruments and optimal timing of assessment before the COS can be confidently implemented. This work lays the groundwork for future refinement and validation efforts that are essential before widespread adoption.

## Supporting information

**S1 File. Comparison of outcomes identified from the systematic review and interviews.**
(PDF)

**S2 File. Rating of outcomes by stakeholder group in round 1 of the Delphi survey.**
(PDF)

**S3 File. Rating of outcomes by stakeholder group in round 2 of the Delphi survey.**
(PDF)

**S4 File. Analysis of attrition bias.**
(PDF)

**S5 File. Rating of outcomes in the consensus meeting.**
(PDF)

## Acknowledgments

We would like to thank all participants who kindly dedicated their time and insights to this study. We would also like to acknowledge the contribution of the palliative care teams at University College London Hospital, Central and North West London NHS Foundation Trust, and the Marie Curie Hospices in Bradford, Hampstead, and the West Midlands, who helped recruit participants for the study.

## Author contributions

**Conceptualization:** Caitlin Medlock, Bella Vivat, Nicola White, Patrick Stone.

**Data curation:** Caitlin Medlock, Bella Vivat, Nicola White, Patrick Stone.

**Formal analysis:** Caitlin Medlock.

**Funding acquisition:** Patrick Stone.

**Investigation:** Caitlin Medlock, Jannicke Rabben.

**Methodology:** Caitlin Medlock, Bella Vivat, Nicola White, Patrick Stone.

**Project administration:** Caitlin Medlock.

**Supervision:** Bella Vivat, Nicola White, Patrick Stone.

**Visualization:** Caitlin Medlock.

**Writing – original draft:** Caitlin Medlock.

**Writing – review & editing:** Caitlin Medlock, Bella Vivat, Nicola White, Jannicke Rabben, Patrick Stone.

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
