## [Decision Letter · Decision Letter 0]

6 Mar 2026

Dear Dr. Medlock,

Thank you for submitting your manuscript to PLOS ONE. After careful consideration, we feel that it has merit but does not fully meet PLOS ONE’s publication criteria as it currently stands. Therefore, we invite you to submit a revised version of the manuscript that addresses the points raised during the review process.

Kindly address all the issues raised by Reviewer 2.

We look forward to receiving your revised manuscript.

Kind regards,

Sylvester Chidi Chima, M.D., L.L.M.

Academic Editor

PLOS One

Journal Requirements:

2. We note that there is identifying data in the Supporting Information file <S1_file.pdf>. Due to the inclusion of these potentially identifying data, we have removed this file from your file inventory. Prior to sharing human research participant data, authors should consult with an ethics committee to ensure data are shared in accordance with participant consent and all applicable local laws.

-Location data

Please remove or anonymize all personal information (Location Data) ensure that the data shared are in accordance with participant consent, and re-upload a fully anonymized data set. Please note that spreadsheet columns with personal information must be removed and not hidden as all hidden columns will appear in the published file.

Reviewers' comments:

Reviewer's Responses to Questions

**Comments to the Author**

1. Is the manuscript technically sound, and do the data support the conclusions?

Reviewer #1: Yes

Reviewer #2: Yes

2. Has the statistical analysis been performed appropriately and rigorously?

Reviewer #1: N/A

Reviewer #2: N/A

3. Have the authors made all data underlying the findings in their manuscript fully available?

Reviewer #1: Yes

Reviewer #2: Yes

4. Is the manuscript presented in an intelligible fashion and written in standard English?

Reviewer #1: Yes

Reviewer #2: Yes

Reviewer #1: A well written manuscript.

The introduction is clear and gives a readable overview of the importance and challenges of prognostication in advanced cancer.

It also gives a cogent case for the need for a core outcome set for the impact of prognostication.

The methods section is clearly written and follows an established methodology. Furthermore, registration on the COMET database ensures this work has not been undertaken by any other group.

The results are clearly presented and understandable.

The discussion is thorough.

I have no major suggestions to improve the manuscript.

It is suitable for publication

Reviewer #2: The manuscript reports on the development of a Core Outcome Set to support the evaluation of the impact on prognostication for people living with advanced cancer, resulting in 9 core outcomes, spanning across 5 domains. This was a detailed, well written manuscript; the design and reporting of findings is robust and clearly articulated, and followed established, standardised methodologies in Core Outcome Set research.

I have a few comments/suggestions.

Clarification on the study advisory group:

An important component of Core Outcome Set consensus studies are the make up of the study advisory group (SAG). In the methods section, ‘phase 3’, in the first paragraph, you touch upon the SAG, but not the make up of the group. I would recommend a brief description of the members/expertise of the study advisory group, including the public partners.

Methods – outcomes and outcome definitions:

You mention you ‘reported on outcomes related to the process or impact of prognostication or explored relevant experience and perceptions’ – how was ‘process or impact’ defined to help steer this? I appreciate there will be more detail in the published findings from the SLR, but I would recommend adding a brief description of these definitions in this manuscript.

Also, do you have a list of the outcome definitions and which outcomes came from which phase (SLR and/or qualitative). This would be a beneficial document/table, perhaps as a supplementary file.

In phase II (qualitative) methods description – while I appreciate it is reported in full elsewhere, a brief sentence or two on how the outcomes were identified/extracted from the qualitative findings would be beneficial here.

PPI, patient/caregiver recruitment numbers:

PPI – any specific detail on how they helped develop the working definitions of outcomes? How did you ensure the terminology was ‘clear, accessible and inclusive to a broader audience’? Were there checked for lay language by public partners?

Consensus meeting – as you have outlined in the manuscript, attrition was more pronounced among patients and caregivers throughout the Delphi, and in the consensus meeting, there was less representation. You very nicely covered attrition and comparison of round 1 results with those that completed both rounds, and I was wondering for the consensus meeting specifically, if you have more information on how you accommodated this smaller representation from patients and caregivers? Did you make any arrangements/accommodations to ensure their voice was heard and they felt confident to participate in the discussions fully during the consensus meeting? Did public partners help with this?

Future work:

This may be beyond the scope of the manuscript, but it is well described that the uptake of a COS is crucial, and many studies cover the ‘what’, but the ‘how’ and ‘when’ is not as often addressed. While you describe future work, are there any definitive plans to carry this on to further validate?

.

Reviewer #1: No

Reviewer #2: No

---

## [Author Response · Author response to Decision Letter 1]

10 Mar 2026

Author response: Thank you for this information, we can confirm that the manuscript adheres to PLOS ONE’s style requirements.

2. We note that there is identifying data in the Supporting Information file <S1_file.pdf>. Due to the inclusion of these potentially identifying data, we have removed this file from your file inventory. Prior to sharing human research participant data, authors should consult with an ethics committee to ensure data are shared in accordance with participant consent and all applicable local laws.

• Name, initials, physical address

• Ages more specific than whole numbers

• Internet protocol (IP) address

• Specific dates (birth dates, death dates, examination dates, etc.)

• Contact information such as phone number or email address

• Location data

• ID numbers that seem specific (long numbers, include initials, titled “Hospital ID”) rather than random (small numbers in numerical order)

Please remove or anonymize all personal information (Location Data) ensure that the data shared are in accordance with participant consent, and re-upload a fully anonymized data set. Please note that spreadsheet columns with personal information must be removed and not hidden as all hidden columns will appear in the published file.

Author response: Thank you for highlighting this. We would like to clarify that the Supporting Information file does not contain any personally identifiable participant data. The table reports only aggregated demographic characteristics, including age ranges (rather than specific ages or dates of birth) and location at the level of country only, with no city, institutional, or other detailed location information provided. No direct identifiers (e.g., names, contact details, addresses, IP addresses, or ID numbers) or specific dates are included. The information presented is therefore not sufficient to identify individual participants and is consistent with our ethical approvals and participant consent. However, if there are specific elements that the editorial team considers should be removed or further anonymised, we would be very happy to revise the file accordingly to ensure full compliance with the journal’s data sharing policy.

Author response: We confirm that the reviewer comments did not include recommendations to cite any specific previously published works. Therefore, no additional citations have been added in response to this point.

Author response: Thank you for this note. We have reviewed the reference list and confirm that none of the cited articles have been retracted. Therefore, no changes to the reference list were required in relation to this point.

Reviewers' comments to the Authors:

Reviewer #1:

A well written manuscript.

The introduction is clear and gives a readable overview of the importance and challenges of prognostication in advanced cancer.

It also gives a cogent case for the need for a core outcome set for the impact of prognostication.

The methods section is clearly written and follows an established methodology. Furthermore, registration on the COMET database ensures this work has not been undertaken by any other group.

The results are clearly presented and understandable.

The discussion is thorough.

I have no major suggestions to improve the manuscript.

It is suitable for publication

Author response: We thank reviewer 1 for their positive comments on the study and the manuscript. We appreciate the reviewer’s assessment that the manuscript is suitable for publication and have no further changes to make in response to these comments.

Reviewer #2:

The manuscript reports on the development of a Core Outcome Set to support the evaluation of the impact on prognostication for people living with advanced cancer, resulting in 9 core outcomes, spanning across 5 domains. This was a detailed, well written manuscript; the design and reporting of findings is robust and clearly articulated, and followed established, standardised methodologies in Core Outcome Set research. I have a few comments/suggestions:

1. Clarification on the study advisory group – An important component of Core Outcome Set consensus studies are the make up of the study advisory group (SAG). In the methods section, ‘phase 3’, in the first paragraph, you touch upon the SAG, but not the make up of the group. I would recommend a brief description of the members/expertise of the study advisory group, including the public partners.

Author response: Thank you for bringing this to our attention. We have revised the manuscript to provide additional detail on the composition of the Study Advisory Group, clarifying that it comprised two people living with advanced cancer, two bereaved informal caregivers, and one current informal caregiver (see lines 281–284).

2. Methods – outcomes and outcome definitions: You mention you ‘reported on outcomes related to the process or impact of prognostication or explored relevant experience and perceptions’ – how was ‘process or impact’ defined to help steer this? I appreciate there will be more detail in the published findings from the SLR, but I would recommend adding a brief description of these definitions in this manuscript.

Author response: Thank you for this helpful suggestion. On reflection, the term process was used incorrectly, as our review focused on the impact of prognostication rather than the process of prognostication itself. In this study, impact refers to the effects of prognostication on patients, informal caregivers, or clinicians. This includes outcomes used to assess the impact of prognostication across a range of domains, including clinical, symptomatic, and psychological outcomes, as well as service-level outcomes such as waiting lists and bed occupancy. We have clarified this in the revised manuscript (see lines 150-152).

3. Also, do you have a list of the outcome definitions, and which outcomes came from which phase (SLR and/or qualitative). This would be a beneficial document/table, perhaps as a supplementary file.

Author response: Thank you for this helpful suggestion. We have now included a supplementary table that presents the outcomes identified from the systematic review and the qualitative interviews, indicating which outcomes were identified in each phase of the study (see Supplementary File 1 and line 176). At this stage of the study, outcomes were not formally defined, as the purpose of these phases was to generate a comprehensive list of potential outcomes to be considered during the subsequent Delphi process and consensus meeting.

4. In phase II (qualitative) methods description – while I appreciate it is reported in full elsewhere, a brief sentence or two on how the outcomes were identified/extracted from the qualitative findings would be beneficial here.

Author response: Thank you for this helpful suggestion. We have now revised the Phase II methods section to explain that outcome identification was undertaken during the framework analysis of the interview transcripts (see lines 170-174). Specifically, transcripts were coded for experiences and perceptions relating to prognostication, with particular attention to consequences or effects that participants described as important. These were then interpreted as potential outcomes, mapped to the modified COMET taxonomy, and compared with those identified in the systematic review to identify any novel outcomes.

5. PPI – any specific detail on how they helped develop the working definitions of outcomes? How did you ensure the terminology was ‘clear, accessible and inclusive to a broader audience’? Were there checked for lay language by public partners?

Author response: Thank you for this helpful question. Working definitions for the outcomes were initially developed and refined through discussion within the Study Advisory Group (SAG), which included public research partners with lived experience of advanced cancer. These discussions helped ensure that the terminology used was clear and accessible. The definitions were then further reviewed and refined within the central research team to ensure consistency and clarity across the outcomes. We did not undertake a separate formal checking process specifically for lay language beyond these discussions. We have aimed to make this clearer in the manuscript (see lines 184-188).

6. Consensus meeting – as you have outlined in the manuscript, attrition was more pronounced among patients and caregivers throughout the Delphi, and in the consensus meeting, there was less representation. You very nicely covered attrition and comparison of round 1 results with those that completed both rounds, and I was wondering for the consensus meeting specifically, if you have more information on how you accommodated this smaller representation from patients and caregivers? Did you make any arrangements/accommodations to ensure their voice was heard and they felt confident to participate in the discussions fully during the consensus meeting? Did public partners help with this?

Author response: Thank you for this important point. We had anticipated equal representation of those with lived experience (patients and informal caregivers) and professionals (clinicians and researchers) at the consensus meeting. However, on the day, fewer patients and informal caregivers attended than expected for a variety of reasons, which limited what could be done at that stage to address the imbalance. We had considered holding separate meetings for patients/informal caregivers and clinicians/researchers, but when asked, patient and informal caregiver participants expressed a preference for a joint meeting. During the meeting, care was taken to ensure that professionals did not dominate the discussion and that patients and informal caregivers had opportunities to contribute, including through use of a round robin approach. In practice, this was less of an issue than anticipated, as those attending were very engaged and outspoken. We have revised the manuscript to make this clearer, including the role of the facilitator and the use of a round robin approach to support balanced discussion and ensure that all participants had equal opportunities to contribute (see lines 217–221).

7. Future work – This may be beyond the scope of the manuscript, but it is well described that the uptake of a COS is crucial, and many studies cover the ‘what’, but the ‘how’ and ‘when’ is not as often addressed. While you describe future work, are there any definitive plans to carry this on to further validate?

Author response: Thank you for this important point. We are currently applying for further funding to support the next phase of this research. This work will focus on further validation and implementation of the COS, including identifying appropriate outcome measurement instruments and exploring strategies to support its uptake in future prognostic impact studies. We have revised the manuscript to make clear that this work is now underway (see line 404).

---

## [Decision Letter · Decision Letter 1]

23 Mar 2026

A Core Outcome Set to evaluate the impact of prognostication in people living with advanced cancer: an international consensus study

PONE-D-25-24643R1

Dear Dr. Medlock,

We’re pleased to inform you that your manuscript has been judged scientifically suitable for publication and will be formally accepted for publication once it meets all outstanding technical requirements.

Kind regards,

Sylvester Chidi Chima, M.D., L.L.M.

Academic Editor

PLOS One

Reviewers' comments:

Reviewer's Responses to Questions

**Comments to the Author**

Reviewer #2: All comments have been addressed

2. Is the manuscript technically sound, and do the data support the conclusions?

Reviewer #2: Yes

3. Has the statistical analysis been performed appropriately and rigorously?

Reviewer #2: Yes

4. Have the authors made all data underlying the findings in their manuscript fully available?

Reviewer #2: Yes

5. Is the manuscript presented in an intelligible fashion and written in standard English?

Reviewer #2: Yes

Reviewer #2: I have nothing to add, the authors addressed each of my points. The manuscript is very thorough and detailed, and it was a fascinating read. It is very suitable for publication. Good luck with the rest of the submission process.

.

Reviewer #2: No

---

## [Editor Report · Acceptance letter]

PONE-D-25-24643R1

PLOS One

Dear Dr. Medlock,

I'm pleased to inform you that your manuscript has been deemed suitable for publication in PLOS One. Congratulations! Your manuscript is now being handed over to our production team.

Kind regards,

on behalf of

Professor Sylvester Chidi Chima

Academic Editor

PLOS One